# Obesity and eating disorders in integrative prevention programmes for adolescents: protocol for a systematic review and meta-analysis

Ana Carolina Barco Leme,[1] Debbe Thompson,[1] Karin Louise Lenz Dunker,[2] Theresa Nicklas,[1] Sonia Tucunduva Philippi,[3] Tabbetha Lopez,[4] Lydi-Anne Vézina-Im,[1] Tom Baranowski[1]

[1]Children's Nutrition Research Center, Baylor College of Medicine, Houston, Texas, USA
[2]Department of Psychiatry, Federal University of São Paulo, São Paulo, Brazil
[3]School of Public Health, University of São Paulo, São Paulo, Brazil
[4]Department of Health and Human Performance, College of Liberal Arts and Social Sciences, University of Houston, Houston, Texas, USA

**Correspondence to**
Dr Ana Carolina Barco Leme; acarol.leme@gmail.com

## ABSTRACT

**Introduction** Obesity and eating disorders are public health problems that have lifelong financial and personal costs and common risk factors, for example, body dissatisfaction, weight teasing and disordered eating. Obesity prevention interventions might lead to the development of an eating disorder since focusing on weight may contribute to excessive concern with diet and weight. Therefore, the proposed research will assess whether integrating obesity and eating disorder prevention procedures ('integrated approach') do better than single approach interventions in preventing obesity among adolescents, and if integrated approaches influence weight-related outcomes.

**Methods and analysis** Integrated obesity and eating disorder prevention interventions will be identified. Randomised controlled trials and quasi-experimental trials reporting data on adolescents ranging from 10 to 19 years of age from both sexes will be included. Outcomes of interest include body composition, unhealthy weight control behaviours and body satisfaction measurements. MEDLINE/PubMed, PsycINFO, Web of Science and SciELO will be searched. Data will be extracted independently by two reviewers using a standardised data extraction form. Trial quality will be assessed using the Cochrane Collaboration criteria. The effects of integrated versus single approach intervention studies will be compared using systematic review procedures. If an adequate number of studies report data on integrated interventions among similar populations (k>5), a meta-analysis with random effects will be conducted. Sensitivity analyses and meta-regression will be performed only if between-study heterogeneity is high (I² ≥75%).

**Ethics and dissemination** Ethics approval will not be required as this is a systematic review of published studies. The findings will be disseminated through conference presentations and peer-reviewed journals.

## BACKGROUND

Paediatric overweight and obesity are worldwide public health concerns,[1] with the highest rates in the USA where 28.8% of boys and 29.7% of girls are overweight or

### Strengths and limitations of this study

► First review and meta-analysis of stand-alone obesity prevention programmes versus integrated obesity and eating disorder prevention approaches on body composition.
► Body composition measures do not precisely measure body fat.
► Disordered eating will be measured using self-reported measures.
► Age will be limited to 10-year-old to 19-year-old adolescents.

obese.[2] Western low-income and middle-income countries (LMIC) also face unhealthy child weight, for example, 24.3% of individuals between 10 and 19 years of age in Brazil were overweight or obese.[2] Some evidence indicates a rapid increase in prevalence levels in LMICs as high or even higher than those found in high-income countries (HICs).[3] Obesity has been associated with long-term and short-term physical health conditions, such as cardiometabolic diseases,[4] certain types of cancers[5] and mental health concerns.[6 7] Overweight youth are also at high risk of becoming obese adults,[8] indicating prevention should be initiated in youth.

Prior systematic reviews have examined childhood obesity prevention studies.[3 9 10] The findings, however, have been mixed. In one review of school-based interventions to prevent obesity among children and adolescents, an average difference between the intervention and control groups was −0.33 kg/m² (−0.55,−0.11, 95% CI), with 84% of this effect explained by the highest quality studies.[11] Alternatively, another reported a difference of 0.03 (95% CI: 0.09 to 0.03, P=0.03) with high heterogeneity (I²=87%).[12] Thus, evidence regarding the effectiveness of

school-based obesity prevention interventions to reduce body mass index (BMI) in youth is mixed with high heterogeneity among studies. Narrower age groups who experience common problems and receive interventions appropriate to these common problems may be more effective.

Eating disorders are illnesses in which the people experience severe disturbances in their eating behaviours and related thoughts and emotions. People with eating disorders typically become preoccupied with food and their body weight.[13] In the Diagnostic and Statistical Manual of Mental Disorders 5, the eating disorders section was renamed 'Feeding and Eating Disorders' and specified three eating disorders: anorexia nervosa, bulimia nervosa and binge eating disorder; and three feeding disorders: pica, rumination disorder and avoidant/restrictive food disorder.[13] These categories and associated criteria served to decrease the frequency of the diagnostic category 'eating disorder not otherwise specified', a heterogeneous not well-defined group of eating disorders. Eating disorder not otherwise specified was the most common diagnosis in clinical and community samples of adolescents, accounting for around 80% of all eating disorder diagnoses, with psychopathology and adverse consequences comparable with anorexia nervosa and bulimia nervosa.[13 14]

Disordered eating behaviours and attitudes are part of the eating disorders continuum and include obsessively thinking about food and calories, becoming angry when hungry, being unable to select what to eat, seeking food to compensate for psychological problems, eating until feeling sick and presenting unreal myths and beliefs about eating and weight.[15] Disordered eating is not limited to those diagnosed with eating disorders. Indeed, many individuals experience disordered eating behaviours, beliefs and feelings towards food but are unaware that they are manifesting 'abnormal' behaviours.[15]

Most interventions in the field of eating disorders can be classified as primary prevention programmes, aiming to reduce risk factors. In general, these interventions focus on girls as a target group based on the observation, that girls have an increased chance of developing an eating disorder, especially anorexia and bulimia nervosa.[16] Schools are the most common setting for the existing evaluated programmes.[16] Earlier eating disorder programmes tended to employ fear appeals, threat appeals or fear arousing communications.[16 17] These methods have been increasingly abandoned, since they did not show an effect or might have even been 'more harmful than beneficial'.[16] More recent eating disorder prevention programmes focused on protective factors such as life skills and emotion regulation competence.[18 19] The PriMa (Primary Prevention of Anorexia Nervosa) programme[16] was a new type of prevention programme for girls up to the age of 12. This scientifically based intervention attempted to prevent eating disorders and reduce disordered eating behaviours by primarily focusing on problems associated with anorexia nervosa. The nine

lesson programme used standardised posters and guidelines to encourage group discussions. The intervention group reported significant improvements in body self-esteem, figure dissatisfaction, knowledge and eating attitudes. Also, instead of interventionists, the programme used school teachers to deliver the intervention.

A recent systematic review and meta-analyses[19] quantified the effectiveness of eating disorder preventive randomised controlled trials for children, adolescents and youth. A total of 112 studies were included; 58% of the trials had high risk of bias. The findings indicated small to moderate effect sizes in reducing eating disorder risk factors. It also revealed that promising preventive interventions for eating disorders risk factors may include cognitive dissonance therapy, cognitive behavioural therapy and media literacy. Whether these interventions lower eating disorder incidence is, however, uncertain, and there is a need for studies that combine eating disorder and obesity prevention.[19]

Although eating disorder prevention programmes included content of relevance to obesity prevention (eg, promotion of healthy weight management), a few assessed the impact on weight status or other obesity-related outcomes.[17 19 20]

Being overweight during childhood increases the chances of having an eating disorder during adulthood (compared with normal weight controls).[16] Common obesity and eating disorders risk factors can be categorised into three levels according to the Social Ecological Model[21]: individual (eg, sex, age and weight status), social (eg, media, weight teasing and ideal beauty pattern) and psychological (eg, self-esteem and body satisfaction). Several studies have described the co-presence of these factors, which could be considered risks for the development of eating disorders and obesity.[22–24] Thus, an integrated approach could address the differences in these prevention philosophies, for example, eating behaviours (dieting vs no dieting) and body weight (lose vs accept weight).[18]

Some interventions addressed both obesity and eating disorders in prevention interventions because of the efficiency in addressing two conditions with a single intervention and a possible reduced risk of inadvertently causing eating disorders while trying to prevent obesity,[25–27] for example, strategies to prevent obesity (monitoring intake and portion control) might unintentionally promote shape concerns and disordered eating. Integrating obesity and eating disorder prevention programmes may prove easier and more cost-effective than treating them separately, and healthy nutrition and physical activity are the focus of both eating disorders and obesity prevention programmes.[28] Body dissatisfaction concerns were also addressed in both approaches, but have mismatched messages. For example, some obesity prevention programmes considered it acceptable to be unhappy about being overweight in order to motivate restricting the amount and content of food consumed to reduce body weight,[28] while eating disorder prevention

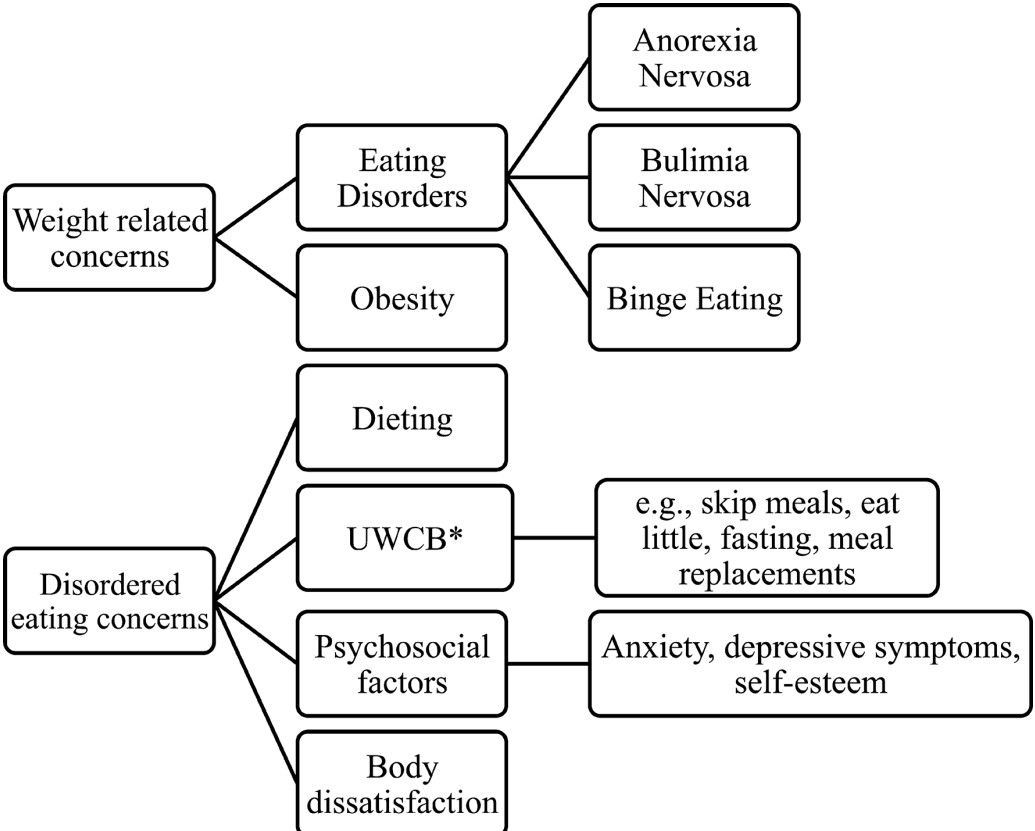

**Figure 1** Scheme of the weight-related behaviours. *UWCB, Unhealthy Weight Control Behaviours.

programmes promoted self-acceptance at any weight, discouraging self-consciousness about dietary intake. However, data supporting these alternative viewpoints are scarce.[28]

Due to the increasing prevalence of obesity and eating disorders[20 25] and shared common risk factors, that is, body dissatisfaction, unhealthy weight control behaviours/ dieting and weight teasing (figure 1), there have been calls for integration to address these common concerns.[25] For instance, obesity and eating disorders can co-occur in the same individual.[25] A cross-cultural comparison between US and Spanish adolescents found dieting and use of unhealthy weight control behaviours were higher among overweight and obese youth and concluded that prevention interventions should address the broad spectrum of eating and weight-related problems.[29]

In summary, obesity and eating disorders have common risk factors with adverse health outcomes, mainly among overweight and female adolescents. Systematic reviews and meta-analyses have only analysed results for single approaches (ie, obesity or eating disorders prevention) and the results have been mixed. Interventions that integrate obesity and eating disorders prevention components might be more effective. A review of such interventions might provide insight into the mechanisms of effect and inform interventions that address both problems simultaneously. To the authors' knowledge, no previous review has identified the impact of integrated obesity and eating disorders prevention programmes for adolescents. The present systematic review will answer the following questions:

► Do integrated obesity and eating disorders interventions do better than obesity-only prevention interventions in improving adolescents' health behaviour outcomes and maintaining healthy weight status?
► Do integrated interventions promote being more satisfied with one's body and reduce unhealthy weight control behaviours in adolescents?

## METHODS AND ANALYSIS
The study protocol was accepted by PROSPERO (www.crd.york.ac.uk/PROSPERO) in October 2017 (CRD42017076547). This protocol follows the Preferred Reporting Items for Systematic Review and Meta-Analysis Protocol (PRISMA-P) checklist.[30] Modifications to the protocol will be tracked and dated in PROSPERO.

## PATIENT AND PUBLIC INVOLVEMENT
Patients and/or public were not involved in this current study.

## CRITERIA FOR CONSIDERING STUDIES FOR THIS REVIEW
### Inclusion criteria
#### Population
Adolescents 10– 19 years of age from both sexes. Adolescents in this age range are at increased risk for unhealthy

weight control behaviours and body satisfaction, shared risk factors for obesity and eating disorders.[18 31] Most published integrated prevention studies are in this age group.

### Types of outcomes

(1) Body composition measurements (ie, body mass index (BMI), waist circumference or per cent body fat); (2) weight control behaviours and/or scales that assess the risk for an eating disorder (such as the Eating Attitudes Test (EAT-26), Sociocultural Attitudes Towards Appearance Questionnaire 3 (SATAQ-3) and Eating Disorder Examination Questionnaire (EDE-Q)); (3) self-reported scales on body satisfaction and (4) other psychological markers (eg, anxiety, depression and/or self-esteem inventories). Inclusion of at least one of the weight control behaviours and/or scales must have been used to assess the risk for eating disorders.[15 18 32 33]

We define 'obesity and eating disorder prevention studies' to be those in which the authors explicitly state they are targeting both sets of outcomes. 'Obesity prevention alone' studies are defined to be those in which the authors state only an obesity prevention objective even if mentioning eating disorder prevention. Some obesity prevention studies collect measures of eating disorders to assess possible unanticipated eating disorder side effects. These will be considered obesity prevention alone studies.

### Study design

Quasi-randomised controlled trials and randomised controlled trials assessing the impact of integrated or obesity-only prevention interventions.

### Types of studies

Quantitative outcome analyses will be included in the systematic review and meta-analysis.

### SEARCH STRATEGY

A structured electronic search will employ all publication years (up to 2018) using four databases and terms will be searched for all text: Medical Literature Library of Medicine (MEDLINE) via PubMed (≥1979), PsycINFO of the American Psychological Association (≥1954), Web of Science via Clarivate Analytics (≥1983) and Scientific Electronic Library (SciELO) via BIREME Latin American and Caribbean Center on Health Science Information (≥1997). Systematic searchers will be developed from this model, applied in MEDLINE: (Obesity) OR Overweight) OR Weight related problems) AND eating disorder) OR weight control behaviors) AND adolescents) OR youth) OR teenagers) OR girls) OR boys) AND prevent*) OR strategies) OR randomized controlled trial. Congress abstracts, dissertations, theses and articles published in journals without peer review will not be included in the review. Only studies written in English, German, Spanish or Portuguese will be included. The results of this search strategy will be reported in a Preferred Reporting Items for Systematic Reviews and Meta-Analyses flow chart.[30]

The bibliographies of papers that match inclusion criteria will be searched by hand to identify further relevant references, which will be subjected to the same screening and selection process. The full search strategy is referred in the online supplementary figure 1.

### Screening and data extraction

All articles identified from the initial electronic search process will be imported into an EndNote library and duplicates are removed. The eligibility criteria will be applied to the results and all identified references are screened independently by two reviewers (ACBL and TL) in a standard blinded way in four stages: (i) reviewing the titles and abstracts; (ii) retrieving and examining the full texts for inclusion; (iii) searching reference lists from the full articles and (iv) examining relevant references for additional studies. TB will be consulted when questions or ambiguity arises. The data extraction form will be pretested with five randomly selected trials.

### Quality assessment

The quality of the randomised controlled trials will be assessed using the Cochrane Collaboration's tool for assessing risk of bias in randomised trials.[34] All data will be extracted and quality assessed by two reviewers. Disagreements at each step will be resolved by discussion. When no consensus is reached, a third reviewer will resolve the discrepancy.

### Data synthesis and analysis

The results of the studies included in the systematic review will be described in a summary table, consisting of author (year), purpose of the study, population targeted, study quality,[34] characteristics of the sample, outcome measures, statistical analyses performed (eg, repeated measures analysis of variance, analysis of covariance or regression analysis) and the results on body composition and disordered eating behaviours. The results of the impact of the intervention will be reported in effect sizes, such as ORs for dichotomous outcomes (eg, satisfied and dissatisfied) or standardised mean differences (SMDs) for continuous outcomes (eg, BMI—kg/m$^2$). All effect sizes will be zero order. To facilitate interpretation and permit comparison with other SMDs and standard effect sizes, the ORs will be converted to Cohen's d.[35] Cohen's d of 0.2 is a small effect size, 0.5 is medium and ≥0.80 is large.[35]

An adequate number of studies (k>5)[36] will trigger a meta-analysis of the findings with a random-effects model. The magnitude of the effect sizes might vary across the studies due to the differences in sample and outcomes of the studies. The pooled effect sizes will be computed, and each study will be weighted according to its sample size. Cochran's Q[37] and I$^2$ statistics[38] will assess the between-study heterogeneity as measures of the percentage of total variation in estimated effects that is a consequence of heterogeneity rather than chance.[39] Significant heterogeneity is considered when the Q statistic has $P<0.05$. An I$^2$ statistic of 25% or less is considered low; 50% moderate

and 75% high heterogeneity.[40] If study heterogeneity exceeds $I^2 \geq 75\%$ (high), it will be explored through sensitivity analyses and meta-regression. The funnel plot will be inspected for publication bias, with a minimum of 10 studies in the analysis,[41] through Duval and Tweedie's trim and fill method[42] and Egger's regression test.[43] All the analyses will be conducted using Comprehensive Meta-Analysis software.

Subgroup analyses might be conducted to assess the possible effects of time differences between integrated prevention versus single obesity approach, and between certain disordered eating behaviours and anthropometric measurements according to the following variables: population (eg, normal weight adolescents vs overweight/obese adolescents) and quality rating (high-rated vs low-rated studies according to the Cochrane Collaboration's tool[34]). Age and sex differences in the impact of integrated prevention programmes will be examined since disorder eating behaviours are more common among older adolescents, girls and overweight/obese individuals.[24 25 44] Moreover, because previous studies[45–48] have found socioeconomic disparities in obesity and disorder eating socioeconomic status, differences will be examined in eating disorder and obesity risk factors in the prevention conditions.

The strength of the evidence will be evaluated using the Grading of Recommendations Assessment, Development and Evaluation (GRADE) guidelines.[49] The following assessments will be made: (1) quality rating for each study according to Cochrane Collaboration's tool[34]; (2) Cohen's d classification to evaluate the magnitude of individual or pooled effect size (SMD),[35] if a meta-analysis is possible; (3) Cochran's $Q$[37] and $I^2$ statistic[38] for heterogeneity and (4) risk of bias by visualising the distribution of the funnel plot if there are at least 10 trials per analysis[41] through Duval and Tweedie's trim and fill method and Eggers's regression test.

## GAPS AND LIMITATIONS

Several gaps and limitations should be noted in anticipation of the findings of the systematic review and meta-analysis. First, the body composition measurements reported in these studies will always be objective measurements (ie, BMI, waist circumference and %body composition) which do not precisely measure percentage body fat.[50–52] Second, eating disorder will be assessed through self-reported measurements which might provide biased responses, since under-reporting is highly prevalent, especially among girls and overweight/obese individuals.[24 47 48 53 54] Any studies that assess disease preventive intervention impacts on self-reported or anthropometric data may potentially underestimate the effect of the intervention.[18] Finally, we are going to cover only adolescents ageing from 10 to 19 years old. However, the majority of the integrated interventions focus on these adolescent years.[18]

## IMPLICATIONS

This systematic review and meta-analysis aims to evaluate the impact of obesity and eating disorder prevention programmes for adolescents. The results of this study should provide new insights into the approaches tested thus far. The systematic review and meta-analysis may also identify specific gaps in the evidence, which would inform the agenda for future research and policy.

## AMENDMENTS

If there is a need to amend this protocol, the date, rationale and a description of each protocol change will be reported.

**Contributors** ACBL, the guarantor of the protocol, drafted the protocol and registered it in PROSPERO. TB reviewed and commented on the protocol in PROSPERO. ACBL, DT, KLLD, TN, STP, TL, L-AV-I and TB all reviewed and commented on this protocol.

**Funding** This work is a publication of the USDA (USDA/ARS - UNITED STATES DEPARTMENT OF AGRICULTURE/AGRICULTURE RESEARCH SERVICE)) Children's Nutrition Research Center, Department of Pediatrics, Baylor College of Medicine (Houston, Texas) and has had been funded, in part, with federal funds from the USDA/ARS under Cooperative Agreement Number 58-6250-6001. ACBL received a postdoctoral fellowship from the State of São Paulo, Brazil (FAPESP - FUNDANÇÃO DE AMPARO À PESQUISA DO ESTADO DE SÃO PAULO - process nº 2016/ 21144-9).

**Competing interests** None declared.

**Patient consent** Not required.

**Ethics approval** Ethics approval will not be required as this is a protocol for systematic review and meta-analysis. This systematic review and meta-analysis will be published in a peer-reviewed journal which will be disseminated electronically and in print.

**Provenance and peer review** Not commissioned; externally peer reviewed.

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
