## [Reviewer comments · BMJ Open]

ARTICLE DETAILS

TITLE (PROVISIONAL)	Obesity and eating disorders in integrative prevention programs for adolescents: protocol for a systematic review and meta-analysis
AUTHORS	Barco Leme, Ana Carolina Thompson, Debbe Lenz Dunker, Karin Nicklas, Theresa Tucunduva Philippi, Sonia Lopez, Tabbetha Vézina-Im, Lydi-Anne Baranowski, Tom

VERSION 1 – REVIEW

REVIEWER	Florian Hammerle University Medical Center of the Johannes Gutenberg University Mainz, Germany
REVIEW RETURNED	22-Nov-2017

GENERAL COMMENTS	This manuscript describes a protocol for a systematic review and meta-analysis of obesity prevention programs versus integrated prevention programs targeting both the prevention of obesity and the development of an Eating Disorder (ED). The authors have been responsive to the raised issues and augmented the manuscript (e.g. the section describing DSM-5 in the Introduction)-thank you. I think the manuscript is substantially improved. In my opinion, there are still some concerns about the manuscript: Reviewing the first version of the manuscript I stated that, “one of my major concerns is the link between obesity prevention and development of an ED. “Older” prevention programs focused on “normal” eating and therefore increased the risk for the development of an ED. In contrast, “newer” prevention programs focus on protective factors as life skills and emotion regulation competence and the risk for the development of an ED seems to be reduced. This devolment of prevention programs should be integrated in both the Background and the Discussion.” I appreciate the meta-regression for time of conduct of the study. The paragraph, lines 28-52 in the Introduction is somewhat augmented but still healthy nutrition and physical activity are stated as common factors of both ED prevention and obesity prevention which seems correct for older programs but not for newer programs; please refer to e.g. Berger et al. (2008) Primary Prevention of Eating Disorders: Characteristics of Effective Programmes and How to Bring Them to Broader Dissemination, Eur. Eat. Disorders Rev. 16, 173–183. I think this should be integrated in the Introduction.
--

	Strengths and Limitations are now included but are somewhat vague. E.g. a strength could be: First review and meta-analysis of stand-alone ED prevention and obesity prevention programs vs. integrated approaches. PRISMA-P and the registration are no real strengths but preconditions. E.g. a limitation might be that the body composition measures do not precisely measure body fat. Please modify this section.
--	---

REVIEWER	Long Khanh-Dao Le Deakin Health Economics, Centre for Population Health Research, School of Health and Social Development, Deakin University, Geelong, Australia.
REVIEW RETURNED	06-Jan-2018

GENERAL COMMENTS	This manuscript is a protocol for systematic review and meta-analysis of integrated prevention programs targeting both the prevention of obesity and eating disorder relative to obesity prevention programs. It is an interesting and important topic. Although the protocol looks good, I think it needs further works as following:  1. Objective: the systematic review aims to compare between active interventions: interventions preventing obesity + ED vs. interventions preventing obesity only. My question is how the authors define 'intervention preventing obesity + ED' and 'interventions preventing obesity only'. For example, if intervention A has the impact on both obesity and ED in one RCT but only on obesity in another RCT, which types of this intervention should be? In addition to this point, why the authors only compare between active interventions and do not compare between active interventions and no intervention control? Please make this point clearly on the manuscript. 2. The authors should consider using more updated references. For example, line 10 to 16 page 5, the cited reference is a systematic review published in 2007. I suggest the authors should use the most recent review for prevention of eating disorder such as "Le LK-D., Barendregt JJ, Hay P, & Mihalopoulos, C. (2017). Prevention of eating disorders: A systematic review and meta-analysis. Clinical Psychology Review, 53, 46–58." Furthermore, this review has touched on prevention of obesity and eating disorder, so the authors should use this article to provide further discussion on current evidence of prevention of obesity and eating disorders. 3. Search term: line 48 to 50 page 6: the authors use DE or ED but I think the authors should consider using full term eating disorder(s). Furthermore, the term 'prevention' should be revised as 'prevent*' to make sure that the authors did not meet any studies. A minor suggestion is that the author needs to search the articles up to 2018 rather than July 2017. Please provide clearly the search terms will be searched for all text or only abstracts. 4. Inclusion criteria: As I understand, the authors will select only RCTs. But it is worthwhile mentioning whether quasi-RCTs are included? Or the RCT studies with school randomisation will be included? 5. Limitation: The authors need to add further limitation such as the review only cover adolescent aging from 10 to 19 so prevention of obesity and eating disorders for other age group remains unclear.
--

VERSION 1 – AUTHOR RESPONSE

Editorial Requests:

- Please improve the 'strengths and limitations' section on page 2. We agree with reviewer 1 that the points presented are currently quite vague. For example, the first point just states the study design. Please make it clearer why each point is a strength or limitation. As a reminder, this section should contain up to five short bullet points, no longer than one sentence each, that relate specifically to the methods of the study reported (see: <http://bmjopen.bmj.com/site/about/guidelines.xhtml#articletypes>). It should not be a summary of the study and its findings.

Response: We thank the editor for comment. We attempted to improve the “strengths and limitations” section on page 2 as requested by reviewer #1. See page 2, line 12-16.

- Please provide a draft of the full search strategy for at least one database as a supplementary file and refer to this in the methods section. Did you consult a librarian/ information scientist when drafting the search strategy?

Response: We added the full search strategy for the database as a supplementary file.

Reviewers comments

Reviewer: 1

Reviewer Name: Florian Hammerle

Institution and Country: University Medical Center of the Johannes Gutenberg University Mainz, Germany

Competing Interests: None declared

This manuscript describes a protocol for a systematic review and meta-analysis of obesity prevention programs versus integrated prevention programs targeting both the prevention of obesity and the development of an Eating Disorder (ED). The authors have been responsive to the raised issues and augmented the manuscript (e.g. the section describing DSM-5 in the Introduction)-thank you. I think the manuscript is substantially improved.

Response: Thank you for your kind comment.

In my opinion, there are still some concerns about the manuscript:

Reviewing the first version of the manuscript I stated that, “one of my major concerns is the link between obesity prevention and development of an ED. “Older” prevention programs focused on “normal” eating and therefore increased the risk for the development of an ED. In contrast, “newer” prevention programs focus on protective factors as life skills and emotion regulation competence and the risk for the development of an ED seems to be reduced. This devolvement of prevention programs should be integrated in both the Background and the Discussion.” I appreciate the meta-regression for time of conduct of the study. The paragraph, lines 28-52 in the Introduction is somewhat augmented but still healthy nutrition and physical activity are stated as common factors of both ED prevention and obesity prevention which seems correct for older programs but not for newer programs; please refer to e.g. Berger et al. (2008) Primary Prevention of Eating Disorders: Characteristics of Effective

Programs and How to Bring Them to Broader Dissemination, *Eur. Eat. Disorders Rev.* 16, 173–183. I think this should be integrated in the Introduction.

Response: We thank you for your suggestion. We have now distinguished between older and newer programs and added the reference mentioned above. See page 4, line 2-13.

Strengths and Limitations are now included but are somewhat vague.

E.g. a strength could be:

First review and meta-analysis of stand-alone ED prevention and obesity prevention programs vs. integrated approaches.

PRISMA-P and the registration are no real strengths but preconditions.

E.g. a limitation might be that the body composition measures do not precisely measure body fat.

Please modify this section.

Response: We added your concerns to the strengths and limitations of our manuscript. See page 2, line 12-16; page 9, line 21-23.

Reviewer: 2

Reviewer Name: Long Khanh-Dao Le

Institution and Country: Deakin Health Economics, Centre for Population Health Research, School of Health and Social Development, Deakin University, Geelong, Australia.

Competing Interests: None declared

This manuscript is a protocol for systematic review and meta-analysis of integrated prevention programs targeting both the prevention of obesity and eating disorder relative to obesity prevention programs. It is an interesting and important topic. Although the protocol looks good, I think it needs further works as following:

1. Objective: the systematic review aims to compare between active interventions: interventions preventing obesity + ED vs. interventions preventing obesity only. My question is how the authors define 'intervention preventing obesity + ED' and 'interventions preventing obesity only'. For example, if intervention A has the impact on both obesity and ED in one RCT but only on obesity in another RCT, which types of this intervention should be? In addition to this point, why the authors only compare between active interventions and do not compare between active interventions and no intervention control? Please make this point clearly on the manuscript.

Response: We appreciate your comment and attempted to make this point clear. See page 6, line 1-3; page 6, line 28-31, page 7, line 1-2.

2. The authors should consider using more updated references. For example, line 10 to 16 page 5, the cited reference is a systematic review published in 2007. I suggest the authors should use the most recent review for prevention of eating disorder such as "Le LK-D., Barendregt JJ, Hay P, & Mihalopoulos, C. (2017). Prevention of eating disorders: A systematic review and meta-analysis. *Clinical Psychology Review*, 53, 46–58.". Furthermore, this review has touched on prevention of obesity and eating disorder, so the authors should use this article to provide further discussion on current evidence of prevention of obesity and eating disorders.

Response: We appreciate your comment and we added the reference. See page 4, line 14-21.

3. Search term: line 48 to 50 page 6: the authors use DE or ED but I think the authors should consider using full term eating disorder(s). Furthermore, the term 'prevention' should be revised as 'prevent*' to make sure that the authors did not meet any studies. A minor suggestion is that the author needs to search the articles up to 2018 rather than July 2017. Please provide clearly the search terms will be searched for all text or only abstracts.

Response: We appreciate your comment and we used the full term eating disorder and made the corrections.

See page 7, line 10-24.

4. Inclusion criteria: As I understand, the authors will select only RCTs. But it is worthwhile mentioning whether quasi-RCTs are included? Or the RCT studies with school randomisation will be included?

Response: We thank you for your comment and we added including quasi-RCT studies as an inclusionary criterion. See page 7, line 3.

5. Limitation: The authors need to add further limitation such as the review only cover adolescent aging from 10 to 19 so prevention of obesity and eating disorders for other age group remains unclear.

Response: We thank you and we added this point as a limitation of our study. See page 9, line 28-29.

VERSION 2 – REVIEW

REVIEWER	Florian Hammerle
REVIEW RETURNED	01-Mar-2018

GENERAL COMMENTS	Thank you for the much improved protocol "Obesity and eating disorders in integrative prevention programs for adolescents: protocol for a systematic review and meta-Analysis". The strengths and limitations and the introduction section describing earlier prevention programs vs. new types of prevention programs is now very much improved-thank you. I also think that the augmentation of the inclusion criteria separating "obesity only"-programs vs. integrated approaches (stated by reviewer #2) improved the manuscript further. The authors have been very responsive to all raised issues and I do not have any further concerns or comments. I wish you a successful systematic review and am looking forward to the results!
---

REVIEWER	Long Khanh-Dao Le
REVIEW RETURNED	26-Feb-2018

GENERAL COMMENTS	The authors have addressed comments.
--------------------------------------